# Immunological Aspects of Richter Syndrome: From Immune Dysfunction to Immunotherapy

**DOI:** 10.3390/cancers15041015

**Published:** 2023-02-05

**Authors:** Abdurraouf Mokhtar Mahmoud, Gianluca Gaidano, Samir Mouhssine

**Affiliations:** Division of Hematology, Department of Translational Medicine, Università del Piemonte Orientale and Azienda Ospedaliero-Universitaria Maggiore della Carità, 28100 Novara, Italy

**Keywords:** Richter syndrome, chronic lymphocytic leukemia, immune dysfunction, immunotherapy

## Abstract

**Simple Summary:**

Richter Syndrome is the development of an aggressive lymphoma in patients affected by chronic lymphocytic leukemia, the most common leukemia in adults. This transformation occurs in 1–10% of chronic lymphocytic leukemia patients and represents an unmet clinical need due to its refractory behavior towards conventional therapies. Recently, the pathogenesis of Richter Syndrome has been shown to be partly related to dysfunction of the immune system, and several immune alterations have been found in patients affected by this disease. Hence, the lack of effective therapies may be overcome through immunotherapy, a type of treatment that improves the immune system response against cancer cells. Several immunotherapeutic approaches have been developed and are currently under investigation, including the use of naked monoclonal antibodies, bispecific antibodies, antibody-drug conjugates, and chimeric antigen receptor-T cells.

**Abstract:**

Richter Syndrome (RS) is defined as the development of an aggressive lymphoma in patients with a previous or simultaneous diagnosis of chronic lymphocytic leukemia (CLL). Two pathological variants of RS are recognized: diffuse large B-cell lymphoma (DLBCL)-type and Hodgkin lymphoma (HL)-type RS. Different molecular mechanisms may explain the pathogenesis of DLBCL-type RS, including genetic lesions, modifications of immune regulators, and B cell receptor (BCR) pathway hyperactivation. Limited data are available for HL-type RS, and its development has been reported to be similar to de novo HL. In this review, we focus on the immune-related pathogenesis and immune system dysfunction of RS, which are linked to BCR over-reactivity, altered function of the immune system due to the underlying CLL, and specific features of the RS tumor microenvironment. The standard of care of this disease consists in chemoimmunotherapy, eventually followed by stem cell transplantation, but limited possibilities are offered to chemo-resistant patients, who represent the majority of RS cases. In order to address this unmet clinical need, several immunotherapeutic approaches have been developed, namely T cell engagement obtained with bispecific antibodies, PD-1/PD-L1 immune checkpoint blockade by the use of monoclonal antibodies, selective drug delivery with antibody-drug conjugates, and targeting malignant cells with anti-CD19 chimeric antigen receptor-T cells.

## 1. Introduction

Richter Syndrome (RS) has been described as the development of an aggressive lymphoma in patients with a previous or simultaneous diagnosis of chronic lymphocytic leukemia (CLL) or small lymphocytic lymphoma (SLL) [1,2,3]. In agreement with the World Health Organization (WHO) classification of Tumours of Haematopoietic and Lymphoid Tissues, two pathological variants of RS are recognized: diffuse large B-cell lymphoma (DLBCL)-type and Hodgkin lymphoma (HL)-type RS [1,2,3,4]. RS is a serious unmet clinical need due to its dismal prognosis, with a median overall survival (OS) of 10 months from the diagnosis [5].

In mature B cells, the variable region of the immunoglobulin heavy chain (IGHV) subunit of the B cell receptor (BCR) results in a particular aminoacidic sequence, different for every B cell clone, that stems from the unique VDJ rearrangement of a given B cell and of its progeny [6]. Approximately 80% of the cases of DLBCL-type RS are clonally related to the CLL phase, as shown by the analysis of the rearrangement of *IGHV* genes, which prove the occurrence of a true transformation event from the previous indolent phase [4,7]. Conversely, ~20% of DLBCL-type RS cases harbor different rearrangements of *IGHV* genes compared to the CLL phase, representing a truly de novo DLBCL development, characterized by a significantly longer median survival (>60 months) [4,8,9].

A typical clinical and laboratory presentation of RS includes B symptoms (weight loss, fever, and night sweats), lymphadenopathy, hypercalcemia, and increased lactate dehydrogenase serum levels [10]. The diagnosis can be achieved through a positive lymph node biopsy performed after a positive ^18^fluorodeoxyglucose positron emission tomography/computed tomography [11].

### 1.1. Morphology of Richter Syndrome

According to the WHO classification, morphologic and phenotypic criteria for the histological diagnosis are analogous to those of classic DLBCL and HL [1]. The histological pattern of DLBCL-type RS consists of an effaced architecture of the lymph node with an infiltrate formed by confluent sheets of large neoplastic post-germinal center B lymphocytes that express CD20, and less frequently CD5 and CD23 [1,4,12]. Compared to de novo DLBCL, where the surface molecule PD-1 is poorly expressed, DLBCL-type RS expresses PD-1 in up to 80% of the cases [13].

HL-type RS shares several analogies with its de novo counterpart, including the presence of CD30^+^/CD15^+^/CD20^−^ Hodgkin and Reed–Sternberg cells, commonly found to be EBV positive [4,14]. Similar to de novo HL, these tumor cells can be found interspersed in a microenvironment formed of other immune cells, such as epithelioid histiocytes, plasma cells, T cells and eosinophils, or, alternatively, surrounded by CLL cells [7,14,15].

### 1.2. Epidemiology

CLL is the most common leukemia in adults in the US, with a median age at diagnosis of 70 years and an incidence of 4.7/100,000 per year [16]. Compared to CLL, de novo DLBCL or HL, RS is undoubtedly a less common condition: both retrospective studies and clinical trials have reported an incidence that ranges between 1 and 10% of CLL patients treated with chemo-immunotherapy (CIT) [17,18,19,20]. An analysis performed by the German CLL Study Group (GCLLSG) has reported a prevalence of 3% in CLL patients treated with CIT alone or with CIT and pathway inhibitors, namely ibrutinib and venetoclax [21].

Although several data on RS incidence in the CIT era have been reported, findings on the epidemiology of RS after treatment with novel agents are limited and seem to be comparable to previous findings [22,23,24]. In first-line treatment of CLL, the first clinical trials with pathway inhibitors have demonstrated an RS incidence similar to that of the CIT era [22,23,24]. Conversely, in relapsed/refractory (R/R) CLL patients, the RS transformation rate was higher compared to the incidence of RS occurrence in patients treated with CIT [25,26,27,28]. These results imply that, for the first-line therapy, pathway inhibitors are not more harmful than CIT, whereas in the context of a R/R disease there is the need for further investigations to test if the apparently higher incidence of RS is explained by the aggressive biological behavior of R/R CLL.

## 2. Immunological Aspects of RS

Different molecular mechanisms explain the pathogenesis of DLBCL-type RS, including genetic lesions, modifications of immune regulators, and BCR pathway hyperactivation [7,10,29]. The main genetic alterations include disruption of *TP53* by deletion or mutation, deletions of *CDKN2A*, gain-of-function mutations of *NOTCH1*, and hyperactivation of *c-MYC* through gene amplification, mutational activation, or chromosomal translocation [7]. Limited data are available for HL-type RS, and its development has been reported to be similar to de novo HL, possibly linked to immunosuppression mediated by Epstein–Barr virus (EBV) infection [7,30,31]. In this review, we will focus on immune-related pathogenesis and dysfunction of the immune system in RS. The main mechanisms involved in the processes are linked to (i) BCR hyperactivation, (ii) altered function of the immune system due to the underlying CLL, and (iii) the characteristics of the tumor RS microenvironment.

### 2.1. BCR Hyperactivation in RS

The BCR is a surface complex formed of an antigen binding subunit, consisting of an immunoglobulin (Ig) molecule, and a signaling subunit, organized in a disulfide-linked heterodimer of the Igα and Igβ proteins, which are non-covalently associated [32]. Inactive BCR can be found spatially isolated or clustered in protein islands on the external membrane of resting B cells [33]. The interaction of the binding subunit with its antigen promotes the clustering of BCR monomeric complexes, the activation of the clustered receptors, and, subsequently, the initiation of the signaling pathway through intracellular protein phosphorylation [33,34].

In the BCR pathway, a pivotal role is played by phosphatidylinositol-3 kinase (PI3K), Bruton tyrosine kinase (BTK), and Akt, which lead to the activation of cell differentiation, proliferation, and survival pathways [35,36]. The hyperactivation of BCR signaling is a major pathogenetic mechanism for the development of CLL and its progression to RS, as demonstrated in vivo in murine xenograft models [37,38,39]. This evidence is corroborated by the results of two different studies, where, on one side, PI3K was found to be constitutively active in all the blood samples of CLL and, on the other side, Akt was hyperactivated in high-risk CLL carrying mutations of *NOTCH1* and *TP53* and in more than 50% of RS biopsies [40,41].

Two mechanisms have been proposed (and probably co-occur) to explain the hyperactivation of BCR signaling in CLL, namely antigen-dependent and antigen-independent engagement of BCR, which may lead to clone selection, proliferation, and survival [39,42]. The antigen-dependent activation is subordinate to the binding of self or exogenous antigens to the BCR, whereas the antigen-independent signaling is initiated by the Ig–Ig interaction within the membrane of the same cell, rather than by activating mutations of the signaling pathway molecules (such as CARD11, CD79a, and CD79b), which appear to be rare [43,44]. Furthermore, ~30% of CLL carries quasi-identical VDJ rearrangement of BCR Ig across different patients, which are groupable in stereotyped subsets, identified by a progressive numbering [45,46]. Remarkably, BCR subset 8, characterized by the IGHV4-39/IGHD6-13/IGHJ5 asset, is frequently associated with peculiar genetic lesions (namely, trisomy 12 and *NOTCH1* mutations) linked to aggressive disease and progression of CLL into DLBCL-type RS, which is more common among patients carrying this specific BCR stereotype [7,47,48]. The association between BCR subset 8 and DLBCL-type RS may be explained by the pronounced reactivity of this BCR subset, which results in a significant CLL cell activation, which, in turn, may lead to the offspring and selection of more aggressive clones [49].

### 2.2. Transformation Mediated by Immune Dysfunction

Immune failure consequent to CLL may represent a major driver for the development of secondary malignances, including RS [50]. In this context, immune dysfunction is caused by defects of both the innate immunity and the adaptive immunity, with hampered function of both lymphoid and myeloid cell lineages [51].

Regarding the lymphoid compartment, several abnormalities have been revealed. CLL cells show unbalanced levels of NK cell inhibitory and activating receptor signals, including abnormally high expression of HLA-E and HLA-G, which, through the interaction with their ligands NKG2A and KIR2DL4 (expressed on NK cells), respectively, cause the suppression of cytokine secretion and inhibit NK cell cytotoxicity against tumor cells [52,53,54]. CLL patients are also characterized by a lower function of γδ T cells, which in normal physiology are able to mount a cytotoxic response against neoplastic cells in a major histocompatibility complex (MHC)-unrestricted mode [55]. In particular, in CLL, γδ T cells are less effective in the induction of tumor cell death through cytokine secretion and degranulation, and, notably, this dysfunction is inducible in healthy γδ cells co-cultured with CLL malignant cells [56].

The most important modifications of the physiological immune response in CLL are represented by major phenotypical and functional CD4^+^ and CD8^+^ T cell alterations, including T cell exhaustion and T cell receptor (TCR) oligoclonality, which drive the impaired immune response [51,57,58].

Furthermore, CLL is characterized by an aberrant expression of immune checkpoint molecules, with a key role played by PD-1 and its ligand PD-L1 [51]. PD-L1 overexpression on the surface of CLL cells downregulates T cell activation and promotes immune tolerance via intracellular pathways through the interaction with PD-1 exposed on the cell membrane of T cells [13,59]. An excessively tolerant immune system is also promoted by the higher circulating T regulatory (Treg) lymphocytes count that characterizes CLL patients [60]. Treg exert their suppressive function through the secretion of regulatory cytokines such as IL-10 and TGF-β, deprivation of co-stimulatory signals to effector cells, and lowering of IL-2 levels. Overall, these immune modifications result in a negative modulation of CD4^+^ and CD8^+^ T cells, NK cells, dendritic cells (DCs), and macrophages [61].

Lastly, EBV may occasionally be involved in the transformation into RS because of the lymphoid immune suppression occurring in CLL, which is intrinsic to the disease and/or may be due to previous chemotherapy [62,63,64]. DLBCL-type RS is found to be EBV-positive less frequently than HL-type RS, which harbor EBV infection in more than 70% of the cases [14,65].

Concerning the myeloid compartment, CLL patients show a defective DC differentiation, driving the emergence of immature DCs that are less efficient in the stimulation of effector T cell response and proliferation [66]. Lastly, circulating myeloid-derived suppressor cells (MDSCs) are present in CLL, which impair effector T cells response and enhance Treg function through various mechanisms, including release of suppressor cytokines, activation of enzymes (e.g., arginase, indoleamine 2,3-dioxygenase, and inducible nitric oxide synthase), and sequestration of cysteine [67,68].

### 2.3. Immune Microenvironment in RS

The immune milieu of RS in lymphoid biopsies is characterized by a significant enhanced expression of PD-1 on neoplastic cells, along with an overexpression of PD-L1 in the surrounding histocytes and DC (Figure 1) [69,70]. Additionally, higher infiltration levels of FOXP3^+^ Treg lymphocytes and CD163^+^ M2-like macrophages were found in the tumor microenvironment of RS compared to that of CLL, suggesting an enhanced immune suppression and resistance against adaptive immunity that promote tumor growth and progression [69,71,72,73]. A classification of tumor microenvironment has been proposed by Teng et al., who divided the tumor milieu into 4 types, according to PD-L1 expression and to the presence or absence of tumor infiltrating lymphocytes (TILs): type I (PD-L1^+^/TIL^+^), leading to adaptive immune resistance; type II (PD-L1^−^/TIL^−^), driving immune ignorance; type III (PD-L1^+^/TIL^−^), characterized by intrinsic oncogenic pathway induction; and type IV PD-L1^−^/TIL^+^, indicating the potential suppressive role of other negative modulators in enhancing immune tolerance [74]. According to this classification, the CLL microenvironment should be classified as type II, thus explaining the relatively poor results obtained with therapies based on PD-1 blockade, whereas the RS microenvironment might fit into a type I definition, therefore justifying the more favorable outcomes of PD-1/PD-L1 inhibition [69,75]. The fact that RS cells overexpress mainly PD-1, while PD-L1 is highly expressed in histiocytes and DCs, seems not to be consistent with the PD-1/PD-L1 model of negative immune modulation described in CLL. Nevertheless, this might be explained by the recent finding of a particular phenotype of PD-1^+^ regulatory B lymphocytes (Bregs) in various cancers (Figure 1) [76,77,78]. The function of PD-1^+^ Breg lymphocytes could be linked to IL-10 production, allowing T cells exhaustion, induction of FOXP3^+^ Treg expansion, and recruitment of MDSCs [79]. Alternatively, these regulatory lymphocytes could inhibit T cell expansion via a PD-L1-dependent pathway, as demonstrated in thyroid tumors [78]. However, the exact mechanism of PD-1^+^ Breg-mediated immune suppression in RS needs to be further investigated.

Another peculiarity of the RS microenvironment is that higher levels of the immune checkpoint molecules LAG3 and TIGIT have been observed in RS, making them potential druggable targets [7,80]. LAG3 is a T cell surface receptor exposed on activated CD4^+^ and CD8^+^ lymphocytes (Figure 1) [81,82]. Although structurally similar to CD4, LAG3 binds with significantly higher affinity to its canonic ligand, namely MHC class II, exposed on the external membrane of antigen presenting cells (APCs) and of various tumor cell types, including DLBCL-type RS cells [72,83,84]. The physiological function of LAG3 is to maintain optimal T cell regulation and homeostasis, but, when challenged by MHC class II expressed on DLBCL-type RS neoplastic cells, it mediates the transmission of inhibitory signals to CD4^+^ and CD8^+^ T cells via intracellular pathways. This results in hampered adaptive immune response against the neoplastic cells due to a decreased function of helper and cytotoxic T lymphocytes, resulting in enhanced tumor escape from apoptosis [72,83].

The immune checkpoint TIGIT is normally expressed on the membrane of NK cells and T lymphocytes and, as a consequence of the interaction with its ligand CD155, a surface antigen physiologically found on APCs, B cells, and T cells, it promotes the downregulation of excessive immune response [85]. On the other side, CD226, a surface receptor which competes with TIGIT for the binding with CD155, mediates activating signaling in T and NK cells [86]. If overactivated, the TIGIT pathway leads to an abnormal intracellular transduction of suppressive stimuli in T and NK cells, thus facilitating neoplastic cell survival in CD155^+^ neoplasms, such as CLL and RS [85,87,88]. A recent study demonstrated that CLL and RS neoplastic cells express TIGIT and CD226, and suggested the existence of an activation and deactivation model analogous to that of T cells also in RS tumor cells, where CD226 might activate neoplastic B cells, enhancing the BCR signaling through the binding with CD155 [87]. In support of this model, RS cells showed an abnormally high level of CD226 compared to TIGIT. This fact implies an imbalance in favor of activating stimuli via CD155–CD226 interaction in RS cells. In contrast, CLL cells displayed higher levels of TIGIT and lower levels of CD226. (Figure 1) [87].

## 3. Immune Therapy Approaches

Due to the poor efficacy reached by standard of care for RS, this disease still represents an unmet medical need and necessitates novel treatment options, including immunotherapy [7]. Different immunotherapy approaches for the treatment of hematological malignancies have made significant progress over the past several years. Advances in understanding the biological factors that suppress an antitumor immune response have led to the development of various antibody-based therapies. This translates in the potential use of immunotherapy for RS treatment, including naked antibodies, bispecific antibodies (bsAbs), antibody-drug conjugates (ADCs), and chimeric antigen receptor (CAR)-T cells (Figure 2) [89]. The currently targeted molecules include surface molecules, namely the surface antigens CD20, CD19, CD30; the receptor tyrosine kinase-like orphan receptor 1 (ROR1); and the immune checkpoint molecules PD-1 and PD-L1 [7]. Evidence on the efficacy of CD37 targeting has been provided by xenograft models, while additional possible targets are represented by recently investigated immune checkpoints, including LAG3, TIGIT, and Casitas B-lineage lymphoma proto-oncogene B (CBL-B) [7,90]. A summary of clinical trials with immunotherapeutic agents in RS can be found in Table 1 (https://clinicaltrials.gov/, last accessed on 7 December 2022).

### 3.1. Standard of Care for RS

Since no randomized clinical trials assessing RS treatment are available, most of the clinical practice is guided by evidence derived from single-arm phase 2 studies (Table 2). The poor efficacy of the CHOP (cyclophosphamide, doxorubicin, vincristine, and prednisone) chemotherapy regimen in RS is slightly improved when combined with rituximab, a human/murine chimeric anti-CD20 monoclonal antibody (mAb) (Table 2) (Figure 2) [91,92]. R-CHOP resulted in an ORR of 67% and a median OS of 21 months, three times longer than CHOP alone [91,92,93,94]. Recent single arm studies incorporating the novel humanized mAb anti-CD20 agent ofatumumab into CHOP (CHOP-O) in substitution of rituximab, and retrospective studies of RS treated with R-EPOCH (rituximab, etoposide, vincristine, cyclophosphamide and doxorubicin), have demonstrated no apparent significant or incremental improvement in outcome with ORR of 46% and OS of 11.4 months for CHOP-O and ORR of 39% and 5.9 months for R-EPOCH (Table 2) (Figure 2) [95]. Remarkably, the addition of the BCL2 inhibitor venetoclax to R-EPOCH allowed better results to be achieved, with an ORR of 62%, an encouraging CR rate of 50%, and a median OS of 19.6 months [96].

Due to the high rate of relapses and poor OS after CIT for patients with RS, consolidation stem cell transplantation (SCT) strategies appear to confer benefit for patients who attain a response to induction therapy [92]. Based on single-center experiences and retrospective analyses, young and fit patients with DLBCL-type RS attaining deep responses with induction treatment can benefit from both autologous SCT and allogeneic SCT [97,98,99,100]. However, only a minority of patients are fit enough or achieve a deep enough response to move on to transplantation, reinforcing the need to identify novel therapeutic strategies that include targeting the unique biological mechanisms that drive RS.

Patients with HL-type RS are commonly treated with conventional HL regimens, with inferior outcomes compared to those reported for de novo HL [101]. The most administered treatment regimen is ABVD (doxorubicin, bleomycin, vinblastine, dacarbazine), which allowed to achieve a median OS of 3–4 years and a median response rate of 20–70% [101,102,103,104].

### 3.2. Immune Checkpoint Blockade

As stated above, PD-1 pathway inhibition appears to play an important therapeutic role in the treatment of certain lymphoma subsets, and RS may also be sensitive to PD-1 blockade [59,69,75,105,106]. Indeed, results of a phase II study (NCT02332980) of pembrolizumab in 25 patients with CLL and RS showed a response in four of nine patients with DLBCL-type RS (ORR = 44%) and a median OS of 10.7 months, suggesting that PD-1 inhibition may be a viable treatment option in this disease (Figure 2) [75].

Based on the synergistic effect of the co-treatment with immune checkpoint inhibitors and BTK inhibitors, it is possible that co-targeting PD-1 and other pathways, especially those linked with BCR signaling, might improve outcomes [107]. For instance, in a phase II trial (NCT02329847) enrolling DLBCL-type RS, the PD-1 inhibitor nivolumab displayed an ORR of 65% (10% complete response, CR) in combination with ibrutinib, a BTK inhibitor [108]. A recently tested nivolumab polytherapy is represented by the combination of the mAb with the PI3Kαδ inhibitor copanlisib, as investigated in an ongoing phase I trial (NCT03884998) [109]. Early reports from the R/R RS cohort of this study suggest a low toxicity profile and a remarkable ORR of 27%, considering that these patients received several prior lines of treatment, including CAR-T cells [109].

The PD-1 pathway may also be blocked by challenging its ligand PD-L1 with the mAb atezolizumab, tested in DLBCL-type RS in an ongoing phase II trial (NCT02846623), combined with the anti-CD20 mAb obinutuzumab and the BCL-2 inhibitor venetoclax [110]. Although the study has already enrolled two different DLBCL-type RS cohorts of patients, treatment-naïve and R/R, respectively, evaluable results have come only from the R/R cohort, for which an encouraging ORR of 100% and a 71% CR rate were reported [110].

Recently, the E3 ubiquitin ligase CBL-B has been investigated as an intracellular immune checkpoint [111]. CBL-B is expressed in various cellular types, including T and NK cells, and has been shown to negatively regulate the activation threshold of T lymphocytes via proteasomal degradation of proteins involved in TCR, CD28, and Notch1 signaling pathways [111,112,113]. The physiological function of CBL-B is to prevent autoimmunity events. Indeed, CBL-B deficient mice display T cell abnormal activation, independent from CD28 costimulatory signal [113]. Additionally, in NK cells, CBL-B promotes the degradation of LAT, an intracellular mediator linked with NK cell activation, resulting in reduced cytotoxic function [114]. For these reasons, CBL-B blockade might increase T and NK cell response against cancer, including RS, and a first-in-human phase I trial, which includes DLBCL-type RS patients treated with the CBL-B inhibitor NX-1607 has been started [90].

While several data on PD-1 blockade in DLBCL-type RS are available, the literature describing this treatment strategy applied to HL-type RS is particularly scarce. Since pembrolizumab has shown notable efficacy in the treatment of de novo R/R HL, a potential therapeutic approach may be represented by this drug [115]. Even though the KEYNOTE-170 (NCT02576990) phase II trial enrolled only two R/R HL-type RS patients, it reported promising efficacy outcomes, with an ORR of 100% (one patient reached a CR and the other achieved a PR) [116].

### 3.3. Bispecific Antibodies

BsAbs are used to describe a large family of molecules designed to bind two different epitopes or antigens [117,118]. The mode of action (MoA) of bsAbs consists in recruiting T cells to kill tumor cells independent of their underlying antigen [119,120]. This novel therapeutic approach has been widely studied and recently approved for the treatment of different B- malignancies (Figure 2) [120,121]. Promising results have been obtained in a monocentric phase II clinical trial (NCT03121534), conducted with the aim of testing efficacy and safety of blinatumomab, a CD19 × CD3 bsAb, in nine patients with RS, of which four showed a reduction in nodal disease, including one CR lasting >1 year [122]. Blinatumomab has been tested also in combination with CIT, as showed in the BLINART study (NCT03931642), a phase II clinical trial, which investigated blinatumomab induction course in DLBCL-type RS after R-CHOP debulking therapy [123]. Blinatumomab treatment, administered only in patients who failed to achieve CR after R-CHOP, demonstrated encouraging anti-tumor activity, with an ORR of 36%, a CR of 20%, and an acceptable toxicity profile [123].

Remarkably, initial favorable results were reached in a phase Ib/II trial (NCT04623541) with the CD20 × CD3 bsAb epcoritamab in DLBCL-type RS previously treated with a first CIT line in nearly half of the cases [124]. Eligible patients, who received subcutaneous epcoritamab monotherapy, achieved an ORR of 60%, with a promising CR rate of 50%. Epcoritamab treatment showed a manageable safety profile, with a low rate of severe hematological treatment-emergent adverse effects (20% anemia and 20% neutropenia) [124]. However, further studies are required to better assess efficacy and toxicity of bsAbs investigating a greater number of patients.

### 3.4. Drug Conjugated Antibodies

An emerging approach for RS treatment is represented by ADCs, molecular complexes formed of a mAb coupled to a cytotoxic payload by a chemical linker (Figure 2) [125]. The MoA of ADCs is to bind to the molecular target of the antibody, usually a surface molecule expressed by the tumor cell, forming then a complex that releases the cytotoxic payload inside the neoplastic cell after its internalization by the cancer cell. This process grants a selective delivery of the toxic compound to tumor cells, causing their death and reducing systemic toxicities [125].

Brentuximab vedotin, widely used for the treatment of de novo R/R HL, is a paradigmatic example of the conjugation of an anti-CD30 mAb and an anti-microtubular agent, namely monomethyl auristatin E (MMAE) [115,126].

ADC therapy for DLBCL-type RS has been investigated with the help of recently developed murine patient-derived xenograft (PDX) models, which have allowed the biology and treatment of this rare disorder to be studied, despite the limited number of human cases [127,128]. The potential efficacy of zilovertamab vedotin (VLS-101), a MMAE-carrying ADC directed against ROR1, has been assessed at a preclinical level using PDX models [129]. ROR1 is a tyrosine kinase receptor located on the surface of normal cells during the embryogenesis, capable of promoting cell survival, proliferation, and migration events via the non-canonical WNT signaling pathway [130]. The majority of normal adult cells, including B cells, do not express ROR1, while it has been found overexpressed in CLL and DLBCL-type RS cells [131,132,133,134]. Mice treated with zilovertamab vedotin did not display treatment related toxicities, showing a remarkable reduction of tumor burden and prolonged survival, with no toxicities recorded, confirming the rationale of a recently started phase II clinical trial (NCT05458297) with this drug [129].

Initial results have come from the use of anti-CD37 mAbs conjugated with the RNA polymerase II inhibitor α-amanitin (amanitin-based ADCs, ATACs) in murine PDX models [135]. CD37 is a surface molecule belonging to the tetraspanin superfamily, selectively highly expressed by normal human B cells, as well as CLL and DLBCL-type RS cells [135,136]. The release of α-amanitin inside the target cells allows the selective fulfillment of its cytotoxic function, interrupting cellular transcription and causing the death of RS cells [135]. Since their efficacy has been tested only in preclinical studies, there is the need for the start of clinical trials with anti-CD37 ATACs in DLBCL-type RS.

Regarding HL-type RS, only two case reports on the use of brentuximab vedotin are present, both describing the treatment of R/R patients, but with contrasting results [137,138]. For this reason, further investigations are needed, possibly based on patient standardization granted by clinical research.

### 3.5. CAR-T Cell Therapy

CD19-directed autologous CAR-T cells have been transformative in the treatment of patients with R/R B cell malignancies, capable of obtaining remarkable response rates and prolonged remissions in this setting.

CAR-T cell therapy is a personalized technology that genetically modifies the patient’s own T lymphocytes to specifically eradicate malignant cells [139]. Key components of commercially available CAR-T cell products consist of an extracellular CD-19 antigen-specific domain, fused with an intracellular domain of a co-stimulatory molecule, such as 4-1BB or CD28, and a CD3ζ signaling domain, which initiates T cell activation signaling and improves CAR-T cell expansion and persistence [140,141]. The process of CAR-T cell therapy includes several steps: leukapheresis, ex vivo genetic engineering and expansion of CAR-T cells, and administration of a lymphodepleting conditioning regimen followed by infusion of CAR-T cell product [139,142].

Partially due to concerns related to CLL-induced immune dysfunction, patients with RS were not included in the pivotal CAR-T cell trials in DLBCL, thus explaining the lack of clinical data regarding the efficacy of anti-CD19 CAR-T [143,144]. Very recently, a retrospective report has analyzed the DESCAR-T registry, which collects data of patients treated with the approved anti-CD19 CAR-T regimens in a real life setting during the last four years, with the aim to obtain evidence on DLBCL-type RS [145]. Considering that most of these patients were heavily pretreated, the encouraging results of the best ORR and best CR rate of 50% and 42%, respectively, were reported, although a higher toxicity was observed compared to de novo DLBCL [145]. Consistently, a recent publication that includes results coming from a phase I clinical trial (NCT03144583) and compassionate use of anti-CD19 ARI-0001 CAR-T cells has documented favorable outcomes for the few DLBCL-type RS patients, reporting an ORR and CR rate of 60% [146]. Highly promising results have also been reached by the experience at James Comprehensive Cancer Center, Ohio, where nine RS patients were treated with axicabtagene-ciloleucel, an anti-CD19 CAR-T therapy, with eight patients reaching an objective response, five of whom achieved a CR [147].

**Table 2 cancers-15-01015-t002:** List of completed phase II clinical trials with chemotherapy and/or CIT in DLBCL-type RS.

Ref	Treatment Regimen	N. RS Patient	Response Rate (%)		Overall Survival (mo)
			ORR	CR	
[93]	R-CHOP	15	67	7	21
[95]	CHOP-O	37	46	27	11.4
[148]	HyperCVXD	29	41	38	10
[149]	Rituximab and GM-CSF with alternating hyperCVXD and MTX/cytarabine	30	43	38	8
[150]	OFAR1	20	50	20	8
[151]	OFAR2	35	39	6.5	6.6
[152]	DHAP, ESHAP	28	43	25	8

Abbreviations: CHOP, cyclophosphamide, doxorubicin, vincristine, and prednisone; CHOP-O, CHOP-ofatumumab; DHAP, dexamethasone, cytarabine, and cisplatin; ESHAP, etoposide, methylprednisolone, cytarabine, and cisplatin; GM-CSF, granulocyte macrophage–colony-stimulating factor; hyper-CVXD, fractionated cyclophosphamide, vincristine, liposomal daunorubicin, and dexamethasone; MTX, methotrexate; OFAR, oxaliplatin, fluradabine, ara-C, and rituximab; R-CHOP, rituximab-CHOP CR; complete response rate; ORR, overall response rate; OS, overall survival.

Overall, CAR-T cell therapy may represent a valid treatment option for RS, but further studies with larger cohorts of patients are needed.

## 4. Conclusions 

In current times, the SoC for RS is represented by CIT, whose poor effectiveness in terms of PFS and OS still defines this lymphoproliferative disorder as an unmet clinical need. The immune-related pathogenesis of RS has been studied to a certain extent, revealing a pivotal role played by BCR hyperreactivity and dysfunction of both innate and adaptive immunity in the previous CLL phase of the disease. Immune microenvironment alterations have been observed in RS, underlining the importance of PD1^+^ Bregs and immune checkpoints in promoting immune tolerance, but additional studies are needed to clarify these crucial pathological aspects. A deeper understanding of the disease has allowed the testing of the efficacy of immunotherapy in clinical trials based on the use of newly developed mAbs (alone or combined with pathway inhibitors) and CAR-T cells, obtaining initial promising results. Overall, there is the need for further investigations, both at the preclinical and clinical sides, in order to fully understand the biology of RS and develop effective treatment strategies, which might include precision-medicine approaches based on individual immune signatures.

## Figures and Tables

**Figure 1 cancers-15-01015-f001:**
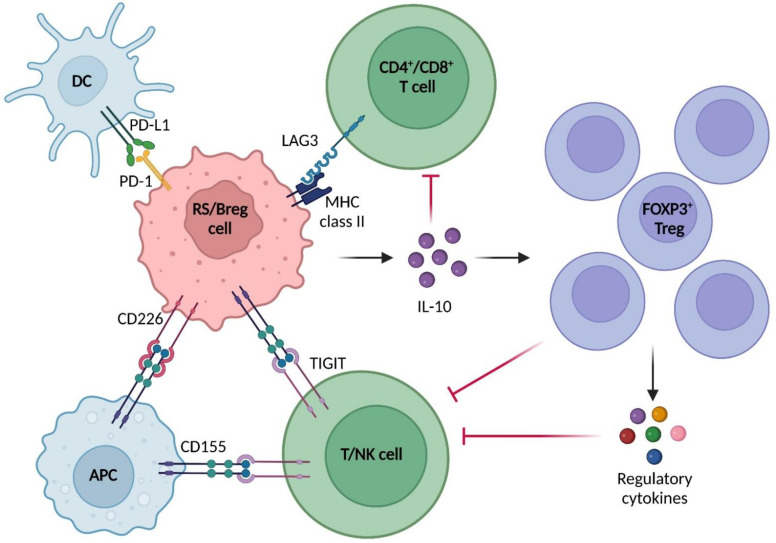
Immune microenvironment interactions in RS. RS tumor cells can act as PD-1^+^ Bregs, promoting T cell exhaustion and immunosuppressive T regulatory lymphocytes expansion via IL-10 secretion or through the interaction with PD-L1 expressed on the surface of DCs and tumor-infiltrating histocytes. Other immune-suppression mechanisms include T cell function inhibition consequent to activation of the immune checkpoints LAG3, found on the cell membrane of T lymphocytes, and TIGIT, on the surface of T and NK cells. Lastly, CD226 hyperexpression and interaction with CD155, expressed on APCs, may act as a positive activator of BCR signaling in RS cells. In a physiological setting, CD 226 acts as a positive activator of T cell response when expressed on the surface of these cells, competing against TIGIT for the binding with CD155. Abbreviations: Breg, B regulatory cell; DC, dendritic cell; Treg, T regulatory cell; APC, antigen presenting cell; RS, Richter syndrome. Image created with Biorender.com (accessed on 7 December 2022).

**Figure 2 cancers-15-01015-f002:**
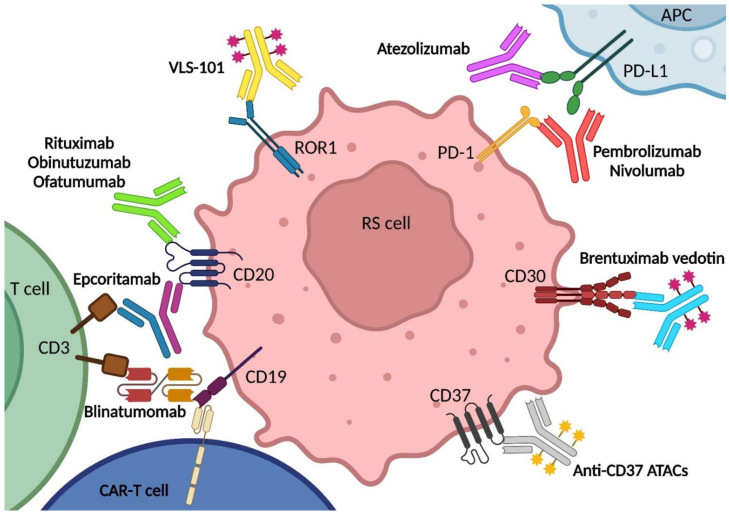
Druggable targets of CIT and immunotherapy in RS. Different CIT and immunotherapy approaches may be used for RS treatment, including chemotherapy in combination with anti-CD20 naked mAbs, PD-1/PD-L1 checkpoint inhibitors in combination with small molecule inhibitors or other mAbs, CD19 × CD3 or CD20 × CD3 bsAbs, anti-CD30, anti-ROR1 or anti-CD37 ADCs, and anti-CD19 CAR-T cells. Abbreviations: APC, antigen presenting cell; CAR, chimeric antigen receptor; ROR1, receptor tyrosine kinase-like orphan receptor 1; ATACs, amanitin-based antibody-drug conjugates; RS, Richter syndrome. Image created with Biorender.com (accessed on 7 December 2022).

**Table 1 cancers-15-01015-t001:** Summary of clinical trials with immunotherapy in RS.

NCT Number	Status	Interventions	Phases
NCT05458297	Recruiting	Single agent: Zilovertamab vedotin	Phase II
NCT05388006	Recruiting	Combination Product: Acalabrutinib with Durvalumab and Venetoclax	Phase II
NCT05025800	Recruiting	Combination Product: CD47 Antagonist ALX148 with Lenalidomide and Rituximab	Phase I/II
NCT04939363	Recruiting	Combination Product: Obinutuzumab with Ibrutinib and Venetoclax	Phase II
NCT04806035	Recruiting	Combination Product: Cosibelimab (TG-1501) plus Ublituximab vs. Cosibelimab (TG-1501) alone	Phase I
NCT04781855	Recruiting	Combination Product: Ibrutinib with Ipilimumab and Nivolumab vs. Ipilimumab plus ibrutinib	Phase I
NCT04679012	Recruiting	Combination Product: Polatuzumab vedotin plus R-EPCH	Phase II
NCT04623541	Recruiting	Single agent: Epcoritamab	Phase I/II
NCT04491370	Recruiting	Autologous SCT followed by Polatuzumab vedotin	Phase I/II
NCT04271956	Recruiting	Combination Product: Tislelizumab plus Zanubrutinib	Phase II
NCT04082897	Recruiting	Combination Product: Obinutuzumabwith Atezolizumab and Venetoclax	Phase II
NCT03884998	Recruiting	Combination Product: Copanlisib plus Nivolumab	Phase I
NCT03778073	Terminated	Combination Product: Cosibelimab (TG-1501) with Ublituximab and Bendamustine	Phase I
NCT03153514	Terminated	Obinutuzumab plus Allogeneic SCT	Phase II
NCT03145480	Terminated	Combination Product: Obinutuzumab plus Ibrutinib vs. Obinutuzumab with Ibrutinib and CHOP	Phase II
NCT03121534	Completed	Single agent: Blinatumomab	Phase II
NCT03113695	Completed	Combination Product: Obinutuzumab with Lenalidomide and HDMP	Phase I
NCT02846623	Recruiting	Combination Product: Atezolizumab with Obinutuzumab and Venetoclax	Phase II
NCT02576990	Completed	Single agent: Pembrolizumab	Phase II
NCT02535286	Completed	Combination Product: Umbralisib with Ublituximab and Cosibelimab (TG-1501)	Phase I
NCT02420912	Completed	Combination Product: Ibrutinib plus Nivolumab	Phase II
NCT03054896	Recruiting	Combination Product: Venetoclax plus DA-EPOCH-R vs Venetoclax plus R-CHOP	Phase II
NCT02332980	Active, not recruiting	Combination product: Pembrolizumab plus Ibrutinib vs. Pembrolizumab plus Idelalisib vs. Pembrolizumab alone	Phase II
NCT02329847	Completed	Combination product: Ibrutinib with Nivolumab	Phase I/II
NCT03931642	Active, not recruiting	Combination product: R-CHOP debulking followed by Blinatumomab	Phase II
NCT01217749	Completed	Combination Product: PCI-32765 plus Ofatumumab	Phase I/II

Abbreviations: R-EPCH, rituximab, etoposide, prednisone, cyclophosphamide, hydroxydaunorubicin; SCT, Stem cell transplantation; CHOP, cyclophosphamide, doxorubicin, vincristine, and prednisone; HDMP, High Dose Methylprednisolone; DA-EPOCH-R, dose-adjusted etoposide, prednisone, vincristine, cyclophosphamide, doxorubicin, and rituximab; R-CHOP, rituximab CHOP.

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
