# Peer review of "Immunological Aspects of Richter Syndrome: From Immune Dysfunction to Immunotherapy"

_cancers, 2023, doi:10.3390/cancers15041015_

Round 1
Reviewer 1 Report
The authors presented an interesting review article about the immunopathology and immunotherapy for Richter Syndrome (RS). However, significant parts of the review were covered by previous reviews, including the author's previous RS review paper and Condoluci et al., 2022 review. This issue negatively impacts the review novelty. It is recommended that the authors refrain from previously published work and present more novel work. The authors should take care of fixing grammatical and punctuation mistakes in the paper.
Author Response
Reviewer 1: The authors presented an interesting review article about the immunopathology and immunotherapy for Richter Syndrome (RS). However, significant parts of the review were covered by previous reviews, including the author's previous RS review paper and Condoluci et al., 2022 review. This issue negatively impacts the review novelty. It is recommended that the authors refrain from previously published work and present more novel work. The authors should take care of fixing grammatical and punctuation mistakes in the paper.
We thank the Reviewer for his/her comments. In order to address the issue raised by the Reviewer, we have added a novel paragraph that focuses on the newly investigated role of Casitas B-lineage lymphoma proto-oncogene B (CBL-B) as an immune checkpoint (lines 370-381). In addition, we have revised the Epidemiology paragraph and excluded most data from previously published reviews. Grammatical and punctuation mistakes have been revised throgout the text.
Reviewer 2 Report
Immunological aspects of Richter Syndrome: from immune dysfunction to immunotherapy
Abdurraouf Mokhtar Mahmoud, Gianluca Gaidano and Samir Mouhssine
Review comments
The review focuses on immune dysregulation and modulation in the CLL patients with Richter transformation. The authors have overviewed a lot of clinical and biological data and tried to deliver updated immunotherapy to readers, which would be beneficial to clinicians to have better understanding of Richter syndrome and therapeutic approaches. Certain aspects not covered in the manuscripts should be included to get it published in the peer reviewed journals.
1. Introduction:
Epidemiology: need to include the not only the incidence of de novo CLL, but also the incidence rate of Richter syndrome, 2-10% of CLL patients. The incidence of de novo DLBCL or HL should be omitted
Morphology: please change to diagnostic criteria of transformation: DLBLC (morphology, phenotype) and HL (morphology and phenotype). If possible, representative images can be added.
The paragraph starting with "since pathway inhibitor…" need to rephrase, emphasizing maybe different transformation rate when using different treatment strategies.
Clinical and laboratory findings at Richter transformation should be included. Also, what is overall outcome when patients' transform to aggressive forms.
2. Immunological aspects of RS:
Before discussing specific mechanisms related to immune dysregulation. Please give a summary of other key mechanisms that could lead to RS, e.g., TP53/del(17p).
DLBCL-type RS is more common than HL-type RS which should be discussed before the latter. Please follow up the rule in each discussion part.
Transformation mediated immune dysfunction:
Please clarify the relationship between decreased CD4/CD8 ratio and occurrence of RS (evidence needed).
Do CLL patients with overexpressing PDL1 show easier RS transformation than those with lower PDL1?
How about the role of EBV infection herein? Relationship of secondary EBV in immune dysregulated CLL patients.
Immune microenvironment in RS: interesting topic. Did the authors have any supporting evidence of different genomic alterations or mutations in the 4 subtypes of tumor milieu?
Figure 1. please more descriptive for the normal function of CD155 and CD226 in the figure legends.
3. Immune therapy approaches: very important for treatment
Please give a summary of the specific receptors in RS status, and explain what mechanisms make them targetable in RS.
Besides chemotherapy R-CHOP vs ABVD, any different immunotherapies for the patients with DLBCL-type RS or HL-RS and response rate? E.g., BsAbs and anti-CD37 are also applied for HL-type RS?
Author Response
Reviewer 2: The review focuses on immune dysregulation and modulation in the CLL patients with Richter transformation. The authors have overviewed a lot of clinical and biological data and tried to deliver updated immunotherapy to readers, which would be beneficial to clinicians to have better understanding of Richter syndrome and therapeutic approaches. Certain aspects not covered in the manuscripts should be included to get it published in the peer reviewed journals.
- Introduction:
Epidemiology: need to include the not only the incidence of de novo CLL, but also the incidence rate of Richter syndrome, 2-10% of CLL patients. The incidence of de novo DLBCL or HL should be omitted
- We thank the Reviewer for his/her constructive comment. We have modified the Epidemiology paragraph accordingly to the suggestions of the Reviewer (lines 78-80).
Morphology: please change to diagnostic criteria of transformation: DLBLC (morphology, phenotype) and HL (morphology and phenotype). If possible, representative images can be added.
- The diagnostic criteria have been changed as suggested and are based on the WHO classification (lines 65-66).
The paragraph starting with "since pathway inhibitor…" need to rephrase, emphasizing maybe different transformation rate when using different treatment strategies.
- We have rephrased the paragraph, emphasizing different transformation rates when using different treatment strategies (lines 90-92).
Clinical and laboratory findings at Richter transformation should be included. Also, what is overall outcome when patients' transform to aggressive forms.
- As suggested by the Reviewer, we have added a sentence on the clinical and laboratory findings typically found in RS (lines 59-63). Since all RS cases are by definition aggressive forms, data on overall outcomes can be found in lines 47-49.
- Immunological aspects of RS:
Before discussing specific mechanisms related to immune dysregulation. Please give a summary of other key mechanisms that could lead to RS, e.g., TP53/del(17p).
- Following the suggestion given by the Reviewer, we have summarized the main genetic lesions which lead to RS (lines 106-109).
DLBCL-type RS is more common than HL-type RS which should be discussed before the latter. Please follow up the rule in each discussion part.
- In agreement with the Reviewer’s suggestion, we have harmonized the order within the discussion (DLBCL-type RS before HL-type RS).
Transformation mediated immune dysfunction:
Please clarify the relationship between decreased CD4/CD8 ratio and occurrence of RS (evidence needed).
- Since the evidence of a relationship between CD4/CD8 ratio and RS is not yet fully established, we feel that the paragraph should be omitted.
Do CLL patients with overexpressing PDL1 show easier RS transformation than those with lower PDL1?
- Although the hypothesis made by the Reviewer is appealing, there are no sufficient data on the correlation between PD-L1 levels before transformation and RS transformation rate. It would be certainly interesting to investigate this topic with scientific studies.
How about the role of EBV infection herein? Relationship of secondary EBV in immune dysregulated CLL patients.
- Information on the status and possible role in transformation of EBV has been added in lines 185-189.
Immune microenvironment in RS: interesting topic. Did the authors have any supporting evidence of different genomic alterations or mutations in the 4 subtypes of tumor milieu?
- Since the tumor milieu classification is currently based on PD-L1 expression, there are no data on genomic alterations or mutations linked to RS microenvironment yet.
Figure 1. please more descriptive for the normal function of CD155 and CD226 in the figure legends.
- As suggested by the Reviewer, we have described the normal function of CD155 and CD226 in figure
- Immune therapy approaches: very important for treatment
Please give a summary of the specific receptors in RS status, and explain what mechanisms make them targetable in RS.
- We have summarized targets and mechanisms of RS immunotherapy, as requested by the Reviewer (lines 273-281).
Besides chemotherapy R-CHOP vs ABVD, any different immunotherapies for the patients with DLBCL-type RS or HL-RS and response rate? E.g., BsAbs and anti-CD37 are also applied for HL-type RS?
Besides the cited treatment regimen, only brentuximab vedotin and pembrolizumab represent a specific immunotherapy for HL-type RS, but data on these treatments are mainly based on case reports (lines 449-452).
Reviewer 3 Report
The review by Mahmoud, Gaidano and Mouhssine represents an excellent and fascinating update of the current knowledge on the immunological aspects of Richter Syndrome. It is written in a very appealing concise style that attracts the reader. All the important immunological aspects of Richter Syndrome are covered in a competent way. The selection of cited and discussed papers is adequate and shows the deep knowledge of the authors about the subject of their research.
For further improvement and completion of this manuscript, I have the following suggestions for modification:
- The Authors describe the function of the immune checkpoint molecules LAG3 and TIGIT and their potential use as druggable targets in RS (page 5). However, the mechanisms of interactions between these molecules on cancer and bystander cells are not clearly described. I suggest describing more clearly those aspects.
- In the manuscript, the order of Table 1 and Table 2 is inverted.
- Table 2 reports a summary of clinical trials with immunotherapy in RS. However, in the text the Authors cite clinical trials (NCT02332980, NCT02329847, NCT03931642) that are not reported in the Table 2. I suggest including them in the Table 2.
Author Response
Reviewer 3: The review by Mahmoud, Gaidano and Mouhssine represents an excellent and fascinating update of the current knowledge on the immunological aspects of Richter Syndrome. It is written in a very appealing concise style that attracts the reader. All the important immunological aspects of Richter Syndrome are covered in a competent way. The selection of cited and discussed papers is adequate and shows the deep knowledge of the authors about the subject of their research.
For further improvement and completion of this manuscript, I have the following suggestions for modification:
- The Authors describe the function of the immune checkpoint molecules LAG3 and TIGIT and their potential use as druggable targets in RS (page 5). However, the mechanisms of interactions between these molecules on cancer and bystander cells are not clearly described. I suggest describing more clearly those aspects.
- We give thanks to the Reviewer for his/her appreciative words in regard of our work. As suggested by the Reviewer, the mechanisms of action of LAG3 and TIGIT have been described in more detail (lines 242-265).
- In the manuscript, the order of Table 1 and Table 2 is inverted.
- The order of table 1 and table 2 has been corrected.
- Table 2 reports a summary of clinical trials with immunotherapy in RS. However, in the text the Authors cite clinical trials (NCT02332980, NCT02329847, NCT03931642) that are not reported in the Table 2. I suggest including them in the Table 2.
- As suggested by the Reviewer, we have integrated the table (former table 2) with the clinical trials (NCT02332980, NCT02329847, NCT03931642).
Round 2
Reviewer 1 Report
The authors presented an interesting review article, addressing specifically the immunological aspects of Richter Syndrome. The current review has a unique and high-quality work that is worth publishing. Proofreading is required for minor grammatical and punctuation mistakes.